# An Investigation of Modular Composable Acoustic Metamaterials with Multiple Nonunique Chambers

**DOI:** 10.3390/ma16247627

**Published:** 2023-12-13

**Authors:** Xiaocui Yang, Xinmin Shen, Daochun Hu, Xiaoyong Wang, Haichao Song, Rongxing Zhao, Chunmei Zhang, Cheng Shen, Mengna Yang

**Affiliations:** 1Engineering Training Center, Nanjing Vocational University of Industry Technology, Nanjing 210023, China; 2018100921@niit.edu.cn (D.H.); 1996100156@niit.edu.cn (X.W.); 2002100250@niit.edu.cn (H.S.); 2019100986@niit.edu.cn (R.Z.); 2019300976@niit.edu.cn (C.Z.); 2016100836@niit.edu.cn (M.Y.); 2MIIT Key Laboratory of Multifunctional Lightweight Materials and Structures (MLMS), Nanjing University of Aeronautics and Astronautics, Nanjing 210016, China; cshen@nuaa.edu.cn; 3Field Engineering College, Army Engineering University of PLA, Nanjing 210007, China; xmshen2014@aeu.edu.cn

**Keywords:** modular composable acoustic metamaterial, noise reduction, multiple nonunique chambers, sound absorption performance, parametric study, sound absorption mechanism

## Abstract

To make the sound absorber easy to fabricate and convenient for practical application, a modular composable acoustic metamaterial with multiple nonunique chambers (MCAM–MNCs) was proposed and investigated, which was divided into a front panel with the same perforated apertures and a rear chamber with a nonunique grouped cavity. Through the acoustic finite element simulation, the parametric studies of the diameter of aperture d, depth of chamber T0, and thickness of panel t0 were conducted, which could tune the sound absorption performances of MCAM–MNCs–1 and MCAM–MNCs–2 for the expected noise reduction effect. The effective sound absorption band of MCAM–MNCs–1 was 556 Hz (773–1329 Hz), 456 Hz (646–1102 Hz), and 387 Hz (564–951 Hz) for T = 30 mm, T = 40 mm, and T = 50 mm, respectively, and the corresponding average sound absorption coefficient was 0.8696, 0.8854, and 0.8916, accordingly, which exhibited excellent noise attenuation performance. The sound absorption mechanism of MCAM–MNCs was investigated by the distributions of the total sound energy density (TSED). The components used to assemble the MCAM–MNCs sample were fabricated by additive manufacturing, and its actual sound absorption coefficients were tested according to the transfer matrix method, which demonstrated its feasibility and promoted its actual application.

## 1. Introduction

Relative to the common sound-absorbing structures and materials, acoustic metamaterials are treated as the most potential acoustic absorbers to reduce noise [1,2], especially in the low- and middle-frequency range, and many kinds of acoustic metamaterials have been developed [3,4,5,6,7,8]. The absorption property of locally reacting acoustic metamaterials with oblique incidence was analyzed by Jiménez et al. [3], which was composed of a slotted panel, each slit being loaded by an array of Helmholtz resonators, and the absorption in the diffuse field took the largest value of 0.951 with an incidence angle around 50.34 degrees. Starkey et al. [5] proposed the thin acoustic metamaterial absorber and consisted only the air and rigid metal, which gave rise to near unity absorption of airborne sound on resonance. Depending on theoretical analysis, an acoustic metamaterial that supported resonance with a monopole (140 Hz) was developed by Gaafer [7] to construct a sound-absorbing technology in low frequency, and the results were of extraordinary correspondence at low frequency and obtained a near-perfect absorption. It was reported by Naimušin and Januševičius [8] that the combination of plastic and rubber structures could be integrated into building structures, which could be utilized as an alternative to reduce noise and reverberation in the field of building acoustics.

There are so many influencing factors to affect the sound absorption properties of the acoustic metamaterials, which indicate that their structural parameters must be optimized to obtain the expected acoustic absorption property [9,10,11,12,13,14,15,16,17,18]. Gurbuz et al. [9] outlined a deep learning-based method to extend the current knowledge of metamaterials and proposed a design method through utilizing conditional generative adversarial networks. Bacigalupo et al. [11] paid particular attention to the optimization of amplitudes and center frequencies of selected stop and pass bands inside the Floquet–Bloch spectra of the acoustic metamaterials featured through the chiral or antichiral microstructure. A way to reduce the total scattering cross-section for a planar configuration of cylinders was proposed by Lai et al. [12] through generative modeling and deep learning. Weeratunge et al. [14] proposed a detailed protocol by coupling machine learning and an optimization algorithm with finite element models, which enabled the inverse and targeted design of underwater acoustic coating. A new machine learning framework to predict the optimal metastructures was developed by Tran et al. [16], such as planar configurations of scatterers with specific functionalities, and the conditional variational autoencoder and supervised variational autoencoder model were proposed. An optimized unit cell design of microslit resonant metamaterial was proposed by de Priester et al. [18] to increase the size of the frequency stopbands and improve the acoustic absorption with normal incidence, and a thorough optimization process of unit cell designs with genetic algorithms was developed, the results of which showed a 9% increase in the first peak sound absorption coefficient compared with the literature standard when the cavity depth was 30 mm and an increase of 10% when the cavity size was 53 mm.

Based on these optimization methods [9,10,11,12,13,14,15,16,17,18], acoustic metamaterials with adjustable sound absorption performances have been developed, which can promote their practical applications in the reduction of noise with variable noise [19,20,21,22]. Yang et al. [19] developed an adjustable parallel Helmholtz acoustic metamaterial to obtain a wide acoustic absorption band by introducing multiple resonant chambers to extend the absorption bandwidth and adjusting the length of the rear cavity for each individual chamber, and the target for all acoustic absorption coefficients above 0.9 was obtained in the frequency range of 602–1287 Hz with the normal incidence and that for all acoustic absorption coefficients above 0.85 was achieved in the frequency range of 618–1482 Hz. An origami-based foldable absorber based on the microperforated resonator was proposed by Jiang et al. [20], and the effective sound absorption was realized via a design whose average thickness was only 1/34.4λ for the resonance frequency. The membrane-type acoustic metamaterials with negative pressure cavities were designed by Xing et al. [21] to obtain a near-perfect acoustic absorption and sound absorption adjustment of the low-frequency spectrum line control, and its advantage was that the locations of acoustic absorption peaks were adjustable. Xu et al. [22] analytically presented and experimentally verified a tunable low-frequency acoustic absorber consisting of multi-layered ring-shaped microslit tubes with a deep subwavelength thickness, and excellent sound absorption (at least 0.9) was obtained in the range of 280–572 Hz in both the simulate data and measurement data. These acoustic metamaterials [19,20,21,22] have the advantages of adjustable functionality, compactness, excellent efficiency, wide-angle sound absorption, and convenient fabrication, which can help pave the way for sound-absorbing metamaterials to be utilized in practical applications in the field of noise reduction.

Therefore, to make the sound absorber easy to fabricate and convenient for practical applications, a modular composable acoustic metamaterial with multiple nonunique chambers (MCAM–MNCs) was proposed and investigated in this study. Its structure was presented and stated, and the theoretical model and finite element simulation model were constructed as well, which provided the foundation for the parametric study. Afterwards, the sample for MCAM–MNCs was prepared through additive manufacturing, and its sound absorption coefficients were tested based on the transfer function method, which verified the feasibility of MCAM–MNCs, the reasonability of the theoretical model, and the effectiveness of the finite element simulation. The sound absorption mechanism of MCAM–MNCs was revealed as well, which could provide references to develop other tunable metamaterials.

## 2. Materials and Methods

### 2.1. Structural Design

The pivotal feature of MCAM–MNCs was multiple nonunique chambers with various sizes, but the individual chambers are not completely independent of each other, and a change in size for one chamber would affect the chambers surrounding it. Therefore, it would be better to divide these chambers into several uniform groups, which was beneficial to decrease the difficulty in design and increase the efficiency of research. Taking the acoustic metamaterial of multiple parallel connection Helmholtz resonators with 16 nonunique chambers as an example, two kinds of MCAM–MNCs were proposed in this study. MCAM–MNCs divided into 4 uniform groups with 4 resonators in each group were labeled MCAM–MNCs–1, as shown in Figure 1a, and that divided into 8 uniform groups with 2 resonators in each group were labeled MCAM–MNCs–2, as shown in Figure 1b. In MCAM–MNCs–1, the whole square metamaterial cell was divided into 4 groups, which were annotated by 4 rectangular frames with different colors in Figure 1a. The size of each group was uniform, and the corresponding rear chamber and front panel were shown in Figure 1c and Figure 1e, respectively. For the convenience of subsequent analysis and explanation, these 16 chambers were labeled C1 to C16, respectively. A similar procedure was performed for MCAM–MNCs–2, and the corresponding rear chamber and the front panel are shown in Figure 1d and Figure 1f, respectively. The width of each chamber, the diameter of each aperture, the thickness of the front aperture, the depth of the rear chamber, and the thickness of each side wall were labeled W_0_, d, t_0_, T_0_, and t, respectively. Meanwhile, the length of each chamber was labeled L_1_ to L_16_, respectively. Taking the following experimental validation by standing wave tube measurement into account, the size of the whole square metamaterial cell was set as 70 mm, as shown in the sectional view of the rear chamber for MCAM–MNCs–1 in Figure 1g and MCAM–MNCs–2 in Figure 1h. Moreover, considering the fabricating cost and bearing capacity, the thickness of side wall t for MCAM–MNCs was selected as 2 mm, which indicated that the width of each chamber W_0_ was 15 mm, as shown in Figure 1g,h. Furthermore, the diameter of each aperture d in the front panel was kept the same, which was beneficial to improve the perforation ratio and enhance the coupling sound absorption effect among the 16 Helmholtz resonators, and the aperture with the same size is conducive to reducing the manufacturing cost and improving the fabricating efficiency as well. The adjustment of sound absorption properties for both MCAM–MNCs–1 and MCAM–MNCs–2 was realized by changing the diameter of each aperture d, the thickness of the front aperture t_0_, and the depth of the rear chamber T_0_.

The relationships among the length of each chamber L_i_ (i = 1, 2, …, 16) for MCAM–MNCs–1 could be expressed by Equation (1), and MCAM–MNCs–2 could be expressed by Equation (2). By this method, the influence of adjustment of each chamber was limited to one group, and it had no effect on the chambers in other groups. Taking the group of chambers C1, C2, C3, and C4 in MCAM–MNCs–1 as an example, the adjustment of L_1_ would affect L_2_, L_3_, and L_4_, and it had no influence on the other 12 chambers. Once more, taking the group of chambers C1 and C2 in MCAM–MNCs–2 as an example, the adjustment of L_1_ would only affect the L_2_, and it had no impact on the other 14 chambers. For the given MCAM–MNCs, by selecting the appropriate series of L_i_ (i = 1, 2, …, 16), the desired acoustic absorption property could be achieved.
(1)$$\left\{\begin{array}{c} L_{1}+L_{2}+L_{3}+L_{4}=4W_{0}\\ L_{5}+L_{6}+L_{7}+L_{8}=4W_{0}\\ L_{9}+L_{10}+L_{11}+L_{12}=4W_{0}\\ L_{13}+L_{14}+L_{15}+L_{16}=4W_{0} \end{array}\right.$$
(2)$$\begin{array}{cc} \left\{\begin{array}{c} L_{1}+L_{2}=2W_{0}\\ L_{3}+L_{4}=2W_{0}\\ L_{5}+L_{6}=2W_{0}\\ L_{7}+L_{8}=2W_{0} \end{array}\right. & \begin{array}{cc} , & \left\{\begin{array}{c} L_{9}+L_{10}=2W_{0}\\ L_{11}+L_{12}=2W_{0}\\ L_{13}+L_{14}=2W_{0}\\ L_{15}+L_{16}=2W_{0} \end{array}\right. \end{array} \end{array}$$

### 2.2. Theoretical Modeling

The theoretical model of the sound absorption coefficient of MCAM–MNCs was constructed on the basement of the Helmholtz resonance principle [23,24,25] with the electro-acoustic theory [26,27,28]. Supposing that the sound absorption coefficient and the acoustic impedance for the proposed MCAM–MNCs were *α* and Z, respectively, the relationship between them could be expressed by Equation (3). Here, *ρ*_0_ was the density of the air, and its value was 1.21 kg/m^3^, and *c*_0_ was the acoustic velocity in air, and its value was 343 m/s.
(3)$$\alpha =1-{\left|\frac{Z-\rho_{0}c_{0}}{Z+\rho_{0}c_{0}}\right|}^{2}$$

MCAM–MNCs consisted of 16 single Helmholtz resonators, which indicated that its acoustic impedance *Z* could be derived through Equation (4), according to the electro-acoustic theory [26,27,28]. Here, *Z_n_* (*n* = 1, 2, …, 16) was the corresponding acoustic impedance for each individual Helmholtz resonator, and it included the acoustic impedance of front aperture *Z_a_* and rear chamber *Z_cn_* (*n* = 1, 2, …, 16), which was shown in Equation (5). The parameters of all these 16 apertures were uniform in this research and had the same diameter of aperture d and length of aperture t_0_ (equal to the thickness of the front panel t_0_), and their acoustic impedances *Z_a_* were the same.
(4)$$Z=\frac{1}{\sum_{n=1}^{16}\left(\frac{1}{Z_{n}}\right)}$$
(5)$$\begin{array}{cc} Z_{n}=Z_{a}+Z_{cn} & n=1,2,\ldots ,16 \end{array}$$

The acoustic impedance of front aperture *Z*_a_ could be calculated by Equation (6) based on the Euler equation [29,30]. Here, ω was the angular frequency of the acoustic wave, and it could be derived by Equation (7) based on normal acoustic frequency *f*; σ was the perforation rate, and it could be derived by Equation (8) according to the diameter of aperture d and the side length of metamaterial cell (4W_0_ + 5t); η was the perforation constant, and it could be derived by Equation (9); *B*_1_(*η*$\sqrt{-i}$) and *B*_0_(*η*$\sqrt{-i}$) were the first order and zero order Bessel functions of the first kind, respectively; and μ was the dynamic viscosity coefficient of air (1.8 × 10^−5^ Pa·s) under the normal conditions. In Equation (8), N was the number of apertures with the same parameters, and its value was 16 in this research. Symbols *W*_0_, *d*, *t*_0_, and *t* were the same as the marked symbols in Figure 1.
(6)$$Z_{a}=i\frac{\omega \rho_{0}t_{0}}{\sigma }{\left[1-\frac{2B_{1}\left(\eta \sqrt{-i}\right)}{\left(\eta \sqrt{-i}\right)\cdot B_{0}\left(\eta \sqrt{-i}\right)}\right]}^{-1}+\frac{\sqrt{2}\mu \eta }{\sigma d}+i\frac{0.85\omega \rho_{0}d}{\sigma }$$
(7)ω=2πf
(8)$$\sigma =\frac{Nd^{2}}{{\left(4W_{0}+4t\right)}^{2}}$$
(9)$$\eta =d\sqrt{\frac{\rho_{0}\omega }{4\mu }}$$

Moreover, the acoustic impedance of the rear chamber *Z_cn_* (*n* = 1, 2, …, 16) was calculated by Equation (10) on the basement of impedance transfer formula [31], and the effective density *ρ*_0en_ and the effective volumetric compressibility c_0en_ of the air domain in the chamber were calculated by Equations (11) and (12), respectively. T_0_ was the uniform depth of the chamber, which was the same as the marked symbol in Figure 1. Furthermore, in Equations (11) and (12), A_n_ was the sectional area of the rectangle chamber, which was the product of length L_n_ and width W_0_, as shown in Equation (13); υ was the kinematic viscosity coefficient of the air under the normal condition, which could be derived by Equation (14); α_xn_ and β_xn_ were 2 intermediate calculation coefficients, which could be calculated by Equations (15) and (16), respectively; and υ’ was the temperature conductivity of the air domain, which could be calculated by Equation (17) based on thermal conductivity κ (0.0258 W/(m·K)) and specific heat capacity C_v_ (718 J/(kg·K)).
(10)$$\begin{array}{cc} Z_{cn}=-i\sqrt{\frac{\rho_{0en}}{C_{0en}}}\cot\left(\omega \sqrt{\rho_{0en}C_{0en}}T_{0}\right) & n=1,2,\ldots ,16 \end{array}$$
(11)$$\begin{array}{cc} \rho_{0en}=\rho_{0}\frac{vA_{n}^{2}}{4i\omega }{\left\{\sum_{x=0}^{\infty }\sum_{y=0}^{\infty }{\left[\alpha_{xn}^{2}\beta_{yn}^{2}\left(\alpha_{xn}^{2}+\beta_{yn}^{2}+\frac{i\omega }{v}\right)\right]}^{-1}\right\}}^{-1} & n=1,2,\ldots ,16 \end{array}$$
(12)$$\begin{array}{cc} C_{0en}=\frac{1}{P_{0}}\left\{1-\frac{4i\omega \left(\gamma -1\right)}{v^{’}A_{n}^{2}}{\sum_{x=0}^{\infty }\sum_{y=0}^{\infty }\left[\alpha_{xn}^{2}\beta_{yn}^{2}\left(\alpha_{xn}^{2}+\beta_{yn}^{2}+\frac{i\omega \gamma }{v^{’}}\right)\right]}^{-1}\right\} & n=1,2,\ldots ,16 \end{array}$$
(13)$$\begin{array}{cc} A_{n}=L_{n}\times W_{0} & n=1,2,\ldots ,16 \end{array}$$
(14)$$v=\frac{\mu }{\rho_{0}}$$
(15)$$\begin{array}{cc} \alpha_{xn}=\frac{\left(x+1/2\right)\pi }{\sqrt{A_{n}}} & n=1,2,\ldots ,16 \end{array}$$
(16)$$\begin{array}{cc} \beta_{yn}=\frac{\left(y+1/2\right)\pi }{\sqrt{A_{n}}} & n=1,2,\ldots ,16 \end{array}$$
(17)$$v^{\prime }=\frac{\kappa }{\rho_{0}C_{v}}$$

In the basement of Equations (3)–(17), the theoretical sound absorption coefficients of MCAM–MNCs for a certain frequency range could be achieved. Afterwards, its prediction accuracy was compared with the acoustic finite element simulation method with the corresponding experimental validation, which could be an effective guide to select the suitable way for the following parametric study and mechanism investigation.

### 2.3. Acoustic Finite Element Simulation

The acoustic finite element simulation model for the proposed MCAM–MNCs was built in the commercial COMSOL Multiphysics 5.5 software, as shown in Figure 2a, which included the perfect matching layer (PML), the background acoustic field (BAF), and the acoustic metamaterial [32,33,34]. The corresponding meshed model for MCAM–MNCs is shown in Figure 2b, and its selected parameters are summarized in Table 1. Similarly, the acoustic finite element simulation model for the single Helmholtz resonator is shown in Figure 2c, and the corresponding structural parameters are labeled in Figure 2d and Figure 2e, respectively. The thickness of the BAF was the same as the single Helmholtz resonator (or acoustic metamaterial), and the PML was 1.5 times greater and could gain an accurate sound absorption coefficient in a certain frequency for the single Helmholtz resonator (or acoustic metamaterial) and ensure the complete absorption of the acoustic wave in the PML to simulate the infinite condition. The acoustic wave with a pressure of 1 Pa was generated in the BAF domain, and the sound absorption coefficient was derived through the calculation of integration for the interfaces between BAF and acoustic metamaterial (or the single Helmholtz resonator). Moreover, regarding the simulation model for the single Helmholtz resonator, the diameter of the BAF domain had a relationship with the aperture and preformation ratio, and it should be equal to $d\cdot \sqrt{N/\sigma }$.

Regarding the acoustic finite element simulation model for the single Helmholtz resonator, the selected parameters were similar to those summarized in Table 1, and the investigated frequency range was 500–1500 Hz to investigate the sound absorption properties of a single Helmholtz resonator with different lengths of the chamber in MCAM–MNCs. The default values for the width of chamber W_0_, the diameter of aperture d, the thickness of front panel t, the depth of chamber T, and the diameter of BAF domain D were 15 mm, 3.5 mm, 2 mm, 30 mm, and 50 mm, respectively, and had taken the size of these 16 single Helmholtz resonators in the proposed MCAM–MNCs into consideration.

### 2.4. Comparative Analaysis

Based on the acoustic finite element simulation model in Figure 1c, the distributions of sound absorption coefficients for a single rectangle Helmholtz resonator with the different lengths of chamber L in the range of 10–24 mm were obtained, as shown in Figure 3, and the corresponding resonance frequencies and peak sound absorption coefficients were summarized in Table 2. It could be found that along with the increase in length in chamber L, the resonance frequency shifted to the low-frequency direction, which was consistent with the common sound absorption principle of a normal Helmholtz resonator [35,36,37]. Meanwhile, it could be observed that the shift was not uniform with the same changing interval of 2 mm for the length of chamber L. The corresponding shift of resonance frequency was 99 Hz along with L changing from 10 mm to 12 mm, while it was only 40 Hz along with L changing from 22 mm to 24 mm, which indicated that the influence of L on resonance frequency was less significant gradually, and the reasonable selection of L for the 16 chambers in MCAM–MNCs was quite important to obtain the desired sound absorption performance. Furthermore, it could be seen in Figure 3 and Table 2 that all these peak sound absorption coefficients were close to 1, which meant that it was very possible to achieve an excellent sound absorption property for MCAM–MNCs by the appropriate combination of multiple rectangle Helmholtz resonators.

The single rectangle Helmholtz resonators were fabricated by a low-force stereolithography (LFS) 3D printer (Formlabs Inc., Summerville, MA, USA) with the length of chamber L = 10 mm, L = 15 mm, and L = 20 mm, respectively, and the corresponding comparisons of sound absorption coefficients in theory, in simulation, and in actual were conducted, as shown in Figure 4. It could be found that the simulation data were closer to the experimental data, no matter whether they were at these resonant sound absorption regions or the non-resonant sound absorption range. The major reason for this phenomenon was that there were many inevitable approximations, reasonable equivalences, ineluctable neglections, and necessary simplifications in the theoretical modeling process, and the acoustic finite element simulation model was closer to the actual situation. Moreover, it could be observed that the deviations between theoretical data and experimental data in the non-resonant sound absorption region were quite remarkable because the theoretical model was constructed mainly for the circumstance of resonant sound absorption, and the non-resonant sound absorption condition was almost not taken into consideration. Therefore, the acoustic finite simulation model instead of the theoretical model was selected to conduct the parametric study and mechanism analysis for MCAM–MNCs in this study.

### 2.5. Parameter Selection for MCAM–MNCs

As shown in Figure 1, there 2 kinds of MCAM–MNCs were investigated in this study. Except for the length of chamber L_i_, the other parameters were the same for MCAM–MNCs–1 and MCAM–MNCs–2, as shown in Table 3 and Table 4. The constraint conditions for these 2 kinds of MCAM–MNCs were shown in Equations (1) and (2), respectively. Regarding the selection of length of chamber L_i_ for MCAM–MNCs–1, the selected values from small to large were 8.5 mm, 9.2 mm, 9.9 mm, 10.6 mm, 11.3 mm, 12.2 mm, 13.1 mm, 14.0 mm, 14.9 mm, 16.0 mm,17.1 mm, 18.2 mm, 19.3 mm, 20.6 mm, 21.9 mm, and 23.2 mm, respectively, the interval of which increased gradually from 0.7 mm to 1.3 mm. The purpose of this selection was to achieve a homogeneous sound absorption performance in the expected frequency range according to the sound absorption principle of a single rectangle Helmholtz resonator with the different lengths of chamber L in Figure 3. According to the constraint conditions in Equation (1), the selected lengths of these 16 chambers from C01 to C16 were reasonably arranged, as shown in Table 3. Similarly, the selected values of the length of chamber L_i_ for MCAM–MNCs–2 from small to large were 5.8 mm, 6.7 mm, 7.7 mm, 8.8 mm, 10.0 mm, 11.3 mm, 12.7 mm, 14.2 mm, 15.8 mm, 17.3 mm, 18.7 mm, 20.0 mm, 21.2 mm, 22.3 mm, 23.3 mm, and 24.2 mm, the interval of which increased gradually from 0.9 mm to 1.6 mm firstly and decreased gradually from 1.6 mm to 0.9 mm because the selections of their values were limited by the constraint conditions in Equation (2). Afterwards, the selected lengths of these 16 chambers from C01 to C16 for MCAM–MNCs–2 were reasonably arranged, as shown in Table 4.

According to the above analysis, it could be found that it was difficult to adjust the sound absorption performance of MCAM–MNCs by changing the selection of lengths of the 16 chambers because there were necessary constraint conditions and a change in one chamber would inevitably affect the adjacent chambers. Thus, for MCAM–MNCs proposed in this study, the lengths of 16 chambers were established, as shown in Table 3 and Table 4, and its sound absorption performance could be tuned by changing the other parameters, such as thickness of panel t_0_, depth of chamber T_0_, and diameter of aperture d.

## 3. Parametric Study

In order to investigate the sound absorption potential of MCAM–MNCs, a parametric study was conducted, and the influences of diameter of aperture d, depth of chamber T_0_, and thickness of panel t_0_ were analyzed one by one. The default parameters were shown in Table 3 and Table 4, the ranges of values for d, T_0_, and t_0_ were 2.0 mm to 3.5 mm with intervals of 0.5 mm, 30 mm, to 50 mm, intervals of 10 mm, and 2.0 mm, to 5.0 mm, and intervals of 1.0 mm, respectively.

### 3.1. The Diameter of Aperture d

The sound absorption performances of MCAM–MNCs with different diameters of aperture d were obtained by the acoustic finite element simulation model in Figure 2, as shown in Figure 5. It could be observed that MCAM–MNCs–1 could gain a relatively homogeneous sound absorption band, and the sound absorption performance of MCAM–MNCs–2 presented the characteristics of high at the front and low at the back in the sound absorption band. The major reason for these phenomena was that the selections of lengths of the sixteen chambers were different in the two kinds of MCAM–MNCs, which had been analyzed in the former section. Meanwhile, along with the increase in the diameter of aperture d, the sound absorption band of MCAM–MNCs shifted to the high-frequency direction, the sound absorption principle of which was the same as the common multiple parallel-connecting Helmholtz resonators [35,36,37]. Moreover, it could be found that the appearance of sound absorption peaks for MCAM–MNCs–2 in Figure 5b was more significant relative to MCAM–MNCs–1 in Figure 5a because the distribution of the length of 16 chambers in MCAM–MNCs–2 was more dispersed than MCAM–MNCs–1, which could be judged from the selected default parameters in Table 3 and Table 4. Furthermore, it could be found that a change in diameter of aperture d would affect the sound absorption property of MCAM–MNCs observably, which indicated that the adjustment of the diameter of aperture d could be considered an efficient method to tune the effective sound absorption band of MCAM–MNCs for the expected noise reduction effect.

### 3.2. The Depth of Chamber T

The sound absorption performances of MCAM–MNCs with different depths of chamber T are shown in Figure 6. It could be observed that along with the increase in the depth in chamber T, both the sound absorption band of MCAM–MNCs–1 and MCAM–MNCs–2 shifted to the low-frequency direction, and the bandwidth of the former was smaller than the latter. Meanwhile, the peak sound absorption coefficients at these resonance frequencies for MCAM–MNCs–1 were larger than those for MCAM–MNCs–2 because the sound absorption effects in the former were more concentrated than those in the latter. Moreover, it could be found that the corresponding peak sound absorption coefficients increased along with the increase in the depth in chamber T, which were quite different from the presented results in Figure 5 with the decrease in the diameter of aperture d because the sound absorption capacity of the former improved through increasing the volume of the rear chamber and the latter, which had almost no variation and the same total thickness in the whole sound absorber. Thus, the change in the depth in chamber T would adjust the sound absorption property of MCAM–MNCs as well, and the corresponding sound absorption capacity of MCAM–MNCs could be improved by increasing its total thickness.

### 3.3. The Thickness of Front Panel t_0_

The sound absorption performances of MCAM–MNCs with the different thicknesses of the front panel t_0_ are shown in Figure 7. It could be observed that the sound absorption curve shifted to the low-frequency direction slightly along with the increase in the thickness in front panel t_0_, and the corresponding total thickness increased accordingly. Relative to the sound absorption characteristics with the change in the diameter of aperture d in Figure 5 and the change in the depth in chamber T in Figure 6, the change in the thickness in the front panel had less impact on the sound absorption performance of MCAM–MNCs, and the total thickness of whole sound absorber increased a bit. Moreover, the total weight of MCAM–MNCs would increase more significantly along with an increase in the thickness in the front panel t_0_ relative to a decrease in the diameter of aperture d and an increase in the depth in chamber T. Thus, the sound absorption performance of MCAM–MNCs could be adjusted mainly through the change in the diameter of aperture d and the depth of chamber T, and it could be finely adjusted by changing the thickness of front panel t_0_ for the expected sound absorption band in a certain frequency range.

It could be observed that the two kinds of MCAM–MNCs had respective advantages. MCAM–MNCs–1 with group number four could obtain a relatively uniform absorption curve, and MCAM–MNCs–2 with group number eight could gain large sound absorption bandwidth, both of which had suitable application scenarios. Moreover, compared with other tunable acoustic metamaterials in the literature [19,20,21,22], the space utilization rate of MCAM–MNCs was higher and could obtain better sound absorption properties with a limited thickness because the space division of the rear chamber could increase the sound absorption efficiency and the front aperture without embedding into the chamber and could improve the acoustic flux. Furthermore, the proposed MCAM–MNCs in this research could achieve an adjustable sound absorption performance as well, and MCAM–MNCs were easier to fabricate and assemble by modular design, which could reduce the manufacturing cost. These advantages made MCAM–MNCs more convenient for practical applications in the field of noise reduction.

Based on the above parametric analysis, it could be found that the width of the effective sound absorption range was determined by the structural parameters and the target frequency band together. For example, when the depth of chamber T increased, the gained sound absorption band would shift to the low-frequency direction, and its width would decrease accordingly, which was consistent with the normal sound absorption principle of the Helmholtz resonator. Analogously, if the target frequency band in the low-frequency range was desired, the structural parameters should be adjusted accordingly to realize the target, and the width of the absorption range would decrease as well. Normally speaking, the wide absorption range was easier to obtain in the high-frequency range and was difficult to achieve in low frequency. Moreover, there were a total of 16 single Helmholtz resonators in MCAM–MNCs, and grouping was necessary to gain effective sound absorption. It could be judged in Figure 5, Figure 6 and Figure 7 that the decrease in group number would enlarge the sound absorption band under the same conditions, but the peak sound absorption coefficients decreased accordingly. Thus, if the uniform sound absorption property in a certain frequency range was needed, MCAM–MNCs–1 with group number four would be better. Otherwise, it would be better to select MCAM–MNCs–2 to gain the sound absorption band as wide as possible. The peak sound absorption coefficient and effective sound absorption band should be taken into consideration simultaneously to select the appropriate group numbers for the proposed MCAM–MNCs in this research.

## 4. Experimental Validation

In order to verify the feasibility of MCAM–MNCs and the reliability of the acoustic finite element simulation model, the components to assemble MCAM–MNCs sample were fabricated by additive manufacturing [38,39,40], and its actual sound absorption coefficients were tested based on the transfer matrix method [41,42,43].

### 4.1. Sample Manufacturing

As shown in Figure 1, the whole MCAM–MNCs could be divided into front panel and rear chamber, which could reduce the manufacturing cost and promote the practical application. The front perforated panel could be fabricated by precision drilling or laser boring for the plate metals or non-metal slabs, which was in favor of rapid mass production. Meanwhile, the rear chamber with cavity could be prepared by precision casting or extrusion forming, which was favorable to realize mass manufacturing. Afterwards, the separately manufactured front panel and rear chamber were assembled to form the proposed MCAM–MNCs. In this research, the components with various parameters were prepared by a Raise3D Pro2 3D printer (Shanghai Fusion Tech Co., Ltd., Shanghai, China) based on the fused filament fabrication method [38,39,40], as shown in Figure 8a–d, which was used to validate the feasibility of MCAM–MNCs. Through the various combinations of the front panel and rear chamber, the sound absorption property in a certain frequency range could be obtained to meet the expected noise reduction effect. Taking MCAM–MNCs–1 with the parameters of d = 3.5 mm, t_0_ = 2 mm, and T = 30 mm as examples, the sample was assembled, as shown in Figure 8e,f. The total thickness of the MCAM–MNCs–1 sample was 34 mm (T + 2 × t_0_ = 34). It should be noted that the perforation in the front panel was not uniform, and each aperture corresponded to 1 cavity in the rear chamber, so the assembly was directional. By choosing the components with different parameters, the sound absorption performance of MCAM–MNCs was tunable.

### 4.2. Standing Wave Tube Testing

The assembled sample of MCAM–MNCs–1 in Figure 8e was tested to gain its actual sound absorption coefficients by an AWA6290T tester (Hangzhou Aihua Instruments Co., Ltd., Hangzhou, China) on the basement of the transfer function method according to the national standard of GB/T 18696.2-2002 (ISO 10534-2:1998) “Acoustics–Determination of sound absorption coefficient and impedance in impedance tubes–part 2: Transfer function method” [41,42,43]. A schematic diagram of standing wave tube testing is shown in Figure 9a. The tested MCAM–MNCs–1 sample was fixed in the sample tube, and its front surface was next to the end of the standing wave tube. The incident sound wave with a certain frequency was generated by the noise generator, amplified by the power amplifier, produced by the sound source, and imported into the standing wave tube. The sound wave reflected by the tested MCAM–MNCs–1 was received by the two microphones installed on the standing wave tube, and the distance between the two microphones and between microphone 2 and the tested MCAM–MNCs–1 sample was 70 mm and 170 mm respectively, which could test the sound absorption coefficients in the frequency range of 200–1600 Hz. The comparisons of sound absorption coefficients of MCAM–MNCs–1 in theory, simulation, and actuality are shown in Figure 9b. It could be observed that the variation tendencies of the three were basically consistent, which could demonstrate the feasibility of proposed MCAM–MNCs, the reasonability of the constructed theoretical model, and the effectiveness of the selected acoustic finite element simulation method. Moreover, it could be seen that the deviation for the theoretical data was larger relative to the simulation data, which was consistent with the comparisons of sound absorption properties in theory, simulation, and experiment for a single Helmholtz resonator in Figure 4. The actual sound absorption coefficients were smaller than the simulation data because there was a fabrication error for the front panel and rear chamber, which would weaken the resonant absorption effect for each Helmholtz resonator and reduce the coupling absorption effect among the different Helmholtz resonators. Furthermore, the actual average sound absorption coefficient of MCAM–MNCs–1 sample in the frequency range of 800–1300 Hz was 0.8511, which showed an extraordinary sound absorption performance, and the sound absorption band and property could be further adjusted by selecting the appropriate structural parameters for the constituent parts to assemble MCAM–MNCs–1. Meanwhile, the first sound absorption peak was gained at the resonance frequency around 830 Hz, which meant that the total thickness of MCAM–MNCs–1 sample 34 mm was only 1/12 of sound wavelength 409 mm (λ = c/f = 340/830 × 1000 = 409 mm). It could be proved that MCAM–MNCs could obtain high absorption efficiency and wide absorption bands simultaneously.

The AWA6290T tester based on the transfer function method could obtain the actual sound absorption coefficients with a normal incidence, which were widely applied in the studies on acoustic metamaterials and other sound-absorbing materials. However, in the practical application, most of the sound waves did not enter the acoustic metamaterial vertically, and the random incidence was the normal situation. Thus, the reverberation chamber method to test the sound absorption performance was closer to the real condition, although it needed more samples and more time. Normally, the sample of acoustic metamaterial was measured through the standing wave tube in the laboratory at the research stage, and it was further tested in the reverberation chamber before practical application.

## 5. Mechanism Investigation

Taking the distributions of the total sound energy density (TSED) around the effective sound absorption band for MCAM–MNCs–1 with the various depths of chamber T as the object, the sound absorption mechanism of MCAM–MNCs was revealed, which was obtained by the acoustic finite element simulation [44,45].

### 5.1. T = 30 mm

The distributions of TSEDs for MCAM–MNCs–1 with the depth of chamber T = 30 mm are shown in Figure 10, in which the investigated frequency points were in the range of 650–1350 Hz with an interval of 50 Hz. In order to make the contrast more reasonable, the scope of data for the legend was limited (−15 × 10^−5^ kg/m^3^, 15 × 10^−5^ kg/m^3^). It could be seen that sound absorption in the effective absorption band was realized by the coupling effect of some Helmholtz resonators, which was judged from the distributions of TSEDs in Figure 10. Taking the sound absorption effect at the frequency point 1150 Hz in Figure 10k as an example, it was realized mainly by the resonance effect of chambers C4 and C12 and assisted by the adjacent chambers C8 and C16. It should be noted that the “adjacent” indicated the chambers with similar sizes instead of the chambers next to each other in space. Homoplastically, the sound absorption effect at the frequency point 1100 Hz in Figure 10j was realized mainly by chambers C3 and C4 and assisted by the adjacent chambers C8, C11, C12, and C16. Meanwhile, it could be judged in Figure 10a,o that beyond the effective sound absorption band 773–1329 Hz (sound absorption coefficient exceeding 0.5), TSEDs were around 0 kg/m^3^ and almost had barely any changes. Therefore, it further proved that the parameters of MCAM–MNCs should be reasonably selected for the expected sound absorption band, which included the diameter of aperture d, the depth of chamber T_0_, and the thickness of panel t_0_ in this study.

### 5.2. T = 40 mm

Similarly, the distributions of TSEDs for MCAM–MNCs–1 with the depth of chamber T = 40 mm are shown in Figure 11, in which the investigated frequency points were in the range of 600–1150 Hz with intervals of 50 Hz. The presented sound absorption mechanism in Figure 11 was similar to Figure 10. The sound absorption effect for certain frequency points was realized by the coupling effect of several Helmholtz resonators. Although the effective sound absorption band was reduced from 773 to 1329 Hz to 646 to 1102 Hz, and the depth of chamber T increased from 30 mm to 40 mm, the average sound absorption coefficient in simulation for the corresponding frequency range was improved from 0.8696 to 0.8854, which indicated that the sound absorption capacity was more concentrated, and it shifted to the low-frequency direction.

### 5.3. T = 50 mm

Analogously, the distributions of TSEDs with the depth of chamber T = 50 mm are shown in Figure 12, in which the investigated frequency points were in the range of 450–1000 Hz with intervals of 50 Hz. It could be observed that the sound absorption effect further shifted to the low-frequency range, and the sound absorption capacity was further concentrated, which was consistent with the calculated effective sound absorption band 556 Hz (773–1329 Hz), 456 Hz (646–1102 Hz), and 387 Hz (564–951 Hz) for T = 30 mm, T = 40 mm, and T = 50 mm, respectively, and the corresponding average sound absorption coefficient was 0.8696, 0.8854, and 0.8916, accordingly. The sound absorption mechanism for MCAM–MNCs was consistent with normal multiple parallel connection Helmholtz resonators.

Moreover, the distributions of acoustic characteristic parameters at the resonance frequency were analyzed as well, which could better explore the sound absorption mechanism of MCAM–MNCs. The distributions of viscous power density, thermal power density and total power density at the resonance frequency of 1203 Hz for MCAM–MNCs–1 with the depth of chamber T = 30 mm are shown in Figure 13. It could be found that the sound absorption in MCAM–MNCs was realized by the thermal viscosity effect, and the viscous energy loss in the front aperture was the dominant factor relative to the thermal energy loss in the rear chamber, because the value of former was thousands of times the latter, which could be judged from the comparisons in Figure 13a,b.

## 6. Conclusions

Through structural design, theoretical modeling, acoustic finite element simulation, and experimental validation, the major achievements gained in this study were as follows.

(1) MCAM–MNCs were divided into a front panel with the same perforated apertures and a rear chamber with nonunique grouped cavities, which made it easy to fabricate and convenient for practical application. By adjusting the parameters of d, T_0_, and t_0_, its sound absorption performance was tunable for the expected noise attenuation effect.

(2) The effective sound absorption band was 556 Hz (773–1329 Hz), 456 Hz (646–1102 Hz), and 387 Hz (564–951 Hz) for T = 30 mm, T = 40 mm, and T = 50 mm, respectively, and the corresponding average sound absorption coefficient was 0.8696, 0.8854, and 0.8916, accordingly, which exhibited an excellent sound absorption performance for MCAM–MNCs.

(3) The sound absorption mechanism of MCAM–MNCs was investigated by the distribution of TSEDs, which proved that the sound absorption was realized by the resonance effect of several Helmholtz resonators and the assistance of adjacent Helmholtz resonators.

The proposed MCAM–MNCs have the advantages of excellent sound absorption performance and adjustable noise reduction frequency bands. They are easy to manufacture and are convenient practical applications, which make them the potential sound absorbers to control the low-frequency variational noise in actual scenarios.

## Figures and Tables

**Figure 1 materials-16-07627-f001:**
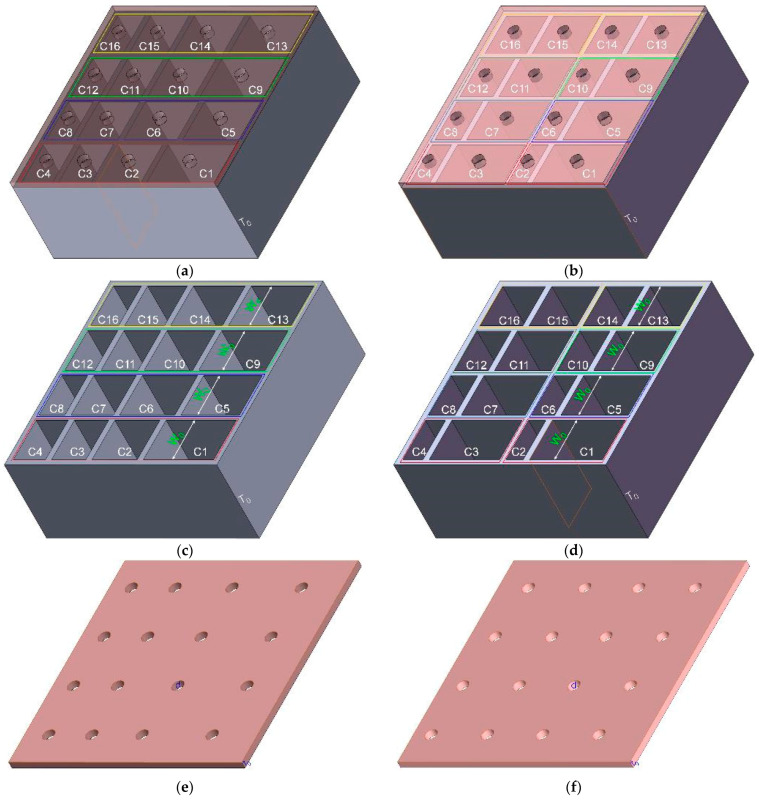
A schematic diagram for MCAM–MNCs. (**a**) MCAM–MNCs–1; (**b**) MCAM–MNCs–2; (**c**) the rear chamber for MCAM–MNCs–1; (**d**) the rear chamber for MCAM–MNCs–2; (**e**) the front panel for MCAM–MNCs–1; (**f**) the front panel for MCAM–MNCs–2; (**g**) the sectional view of the rear chamber for MCAM–MNCs–1; and (**h**) the sectional view of the rear chamber for MCAM–MNCs–2.

**Figure 2 materials-16-07627-f002:**
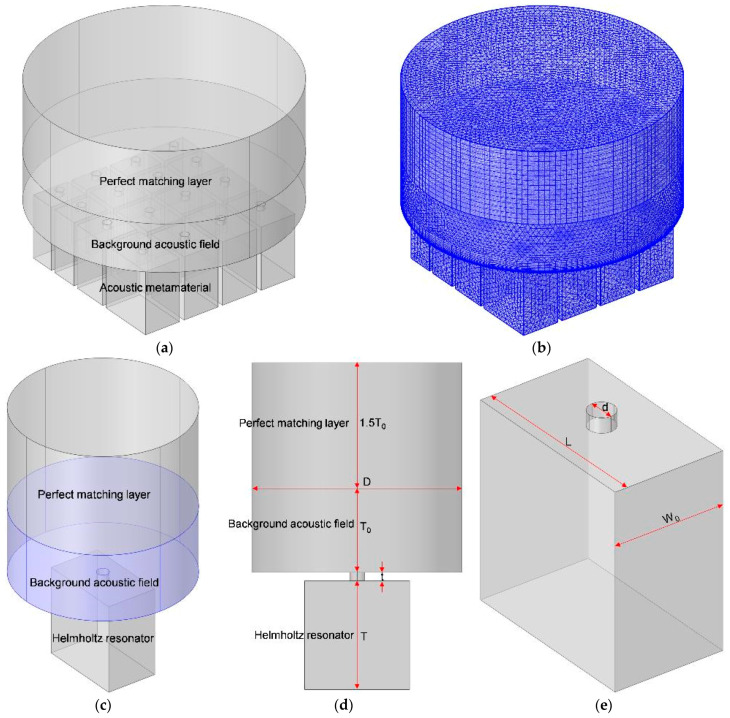
The acoustic finite element simulation models. (**a**) The model for MCAM–MNCs; (**b**) the corresponding meshed model for (**a**); (**c**) the model for the single Helmholtz resonator; (**d**) the corresponding front view of (**c**); and (**e**) a schematic diagram of the single Helmholtz resonator.

**Figure 3 materials-16-07627-f003:**
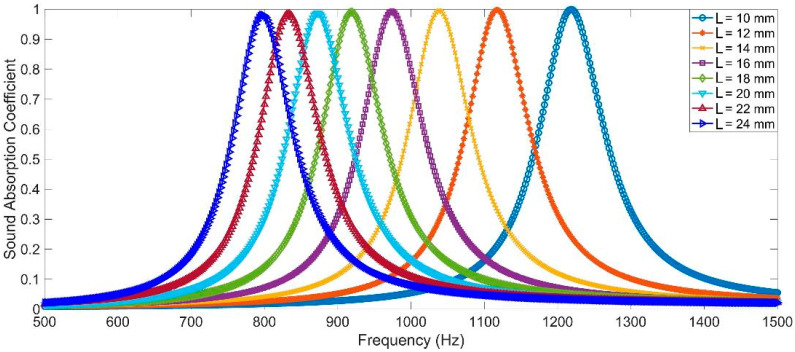
The distributions of sound absorption coefficients for a single rectangle Helmholtz resonator with the different lengths of chamber L in the range of 10–24 mm.

**Figure 4 materials-16-07627-f004:**
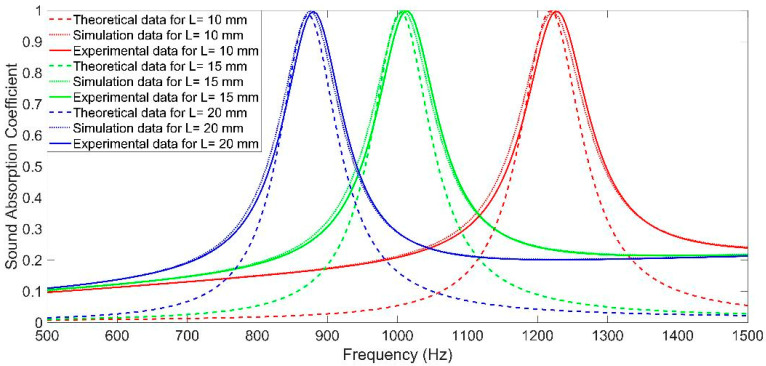
Comparisons of the sound absorption properties in theory, simulation, and experiment for a single Helmholtz resonator when the length of chamber L was 10 mm, 15 mm, and 20 mm.

**Figure 5 materials-16-07627-f005:**
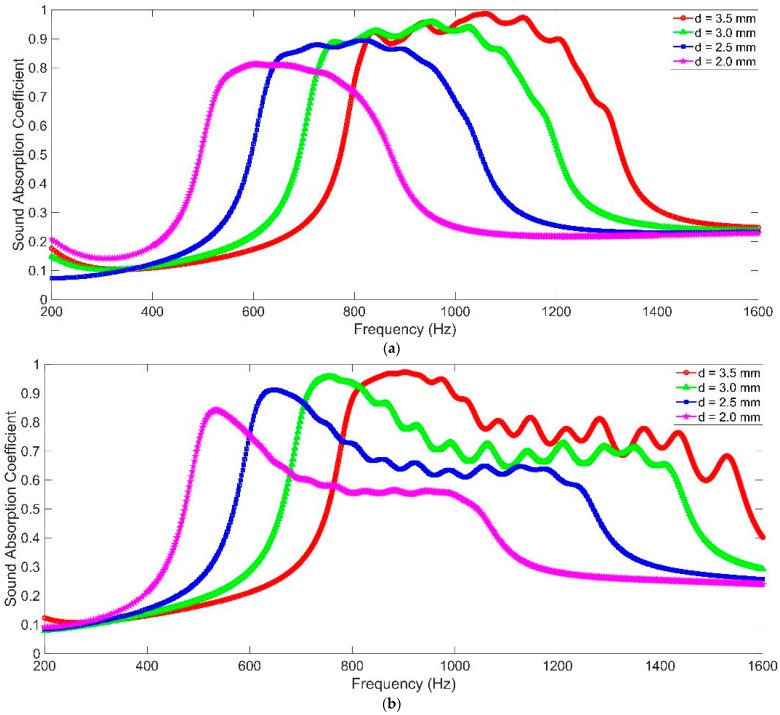
The sound absorption performance of MCAM–MNCs with different diameters of aperture d. (**a**) MCAM–MNCs–1; and (**b**) MCAM–MNCs–2.

**Figure 6 materials-16-07627-f006:**
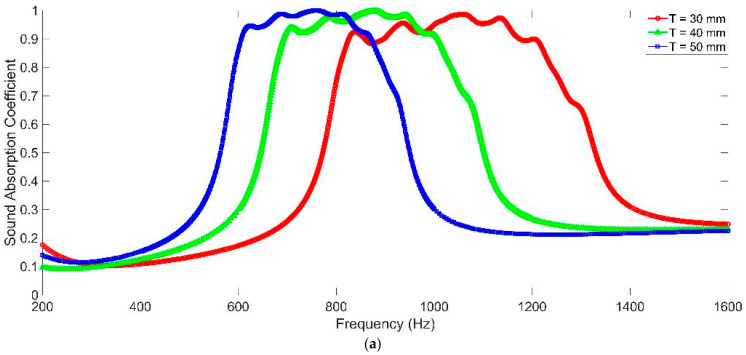
The sound absorption performance of MCAM–MNCs with different depths of chamber T. (**a**) MCAM–MNCs–1; and (**b**) MCAM–MNCs–2.

**Figure 7 materials-16-07627-f007:**
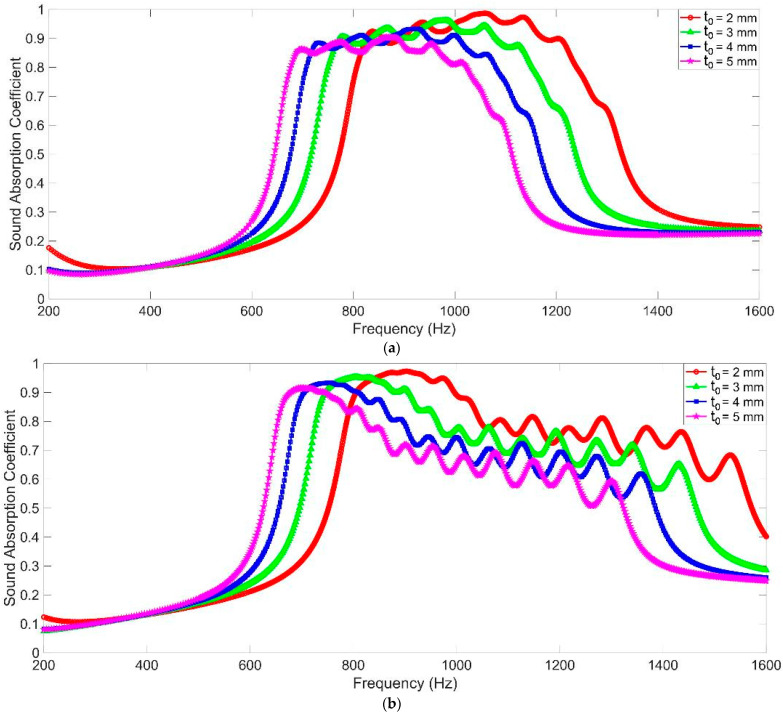
The sound absorption performance of MCAM–MNCs with different thicknesses of front panel t_0_. (**a**) MCAM–MNCs–1; and (**b**) MCAM–MNCs–2.

**Figure 8 materials-16-07627-f008:**
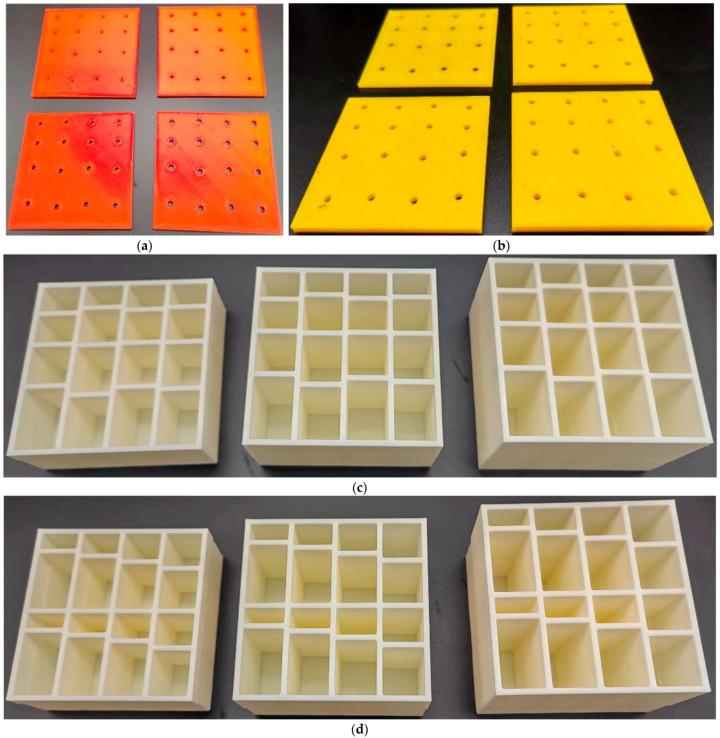
The actual samples prepared by additive manufacturing. (**a**) The front panels with various diameters of aperture d; (**b**) the front panels with various thicknesses t_0_; (**c**) the chamber for MCAM–MNCs–1; (**d**) the chamber for MCAM–MNCs–2; (**e**) the assembled sample of MCAM–MNCs–1; and (**f**) the second view for the assembled sample of MCAM–MNCs–1.

**Figure 9 materials-16-07627-f009:**
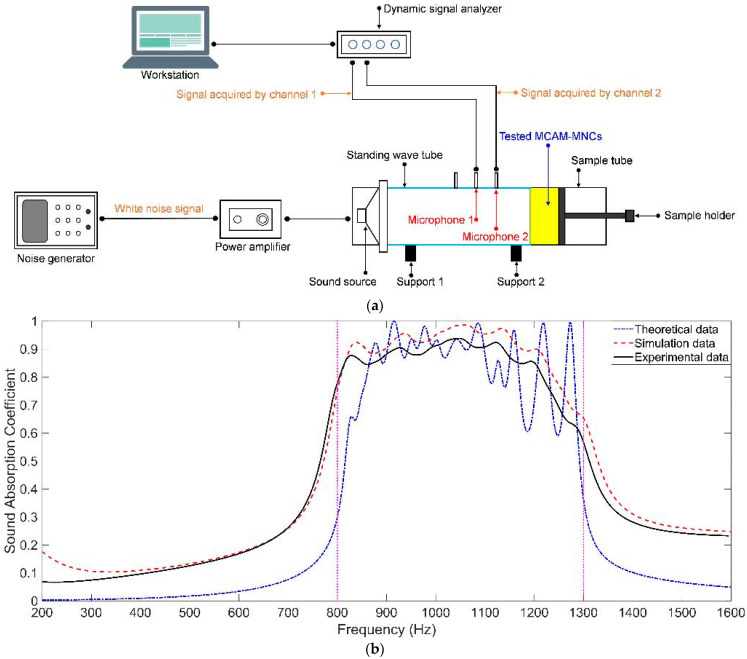
The experimental validation of MCAM–MNCs–1. (**a**) A schematic diagram of standing wave tube testing; and (**b**) comparisons of the sound absorption coefficients of MCAM–MNCs–1 in theory, simulation, and actuality.

**Figure 10 materials-16-07627-f010:**
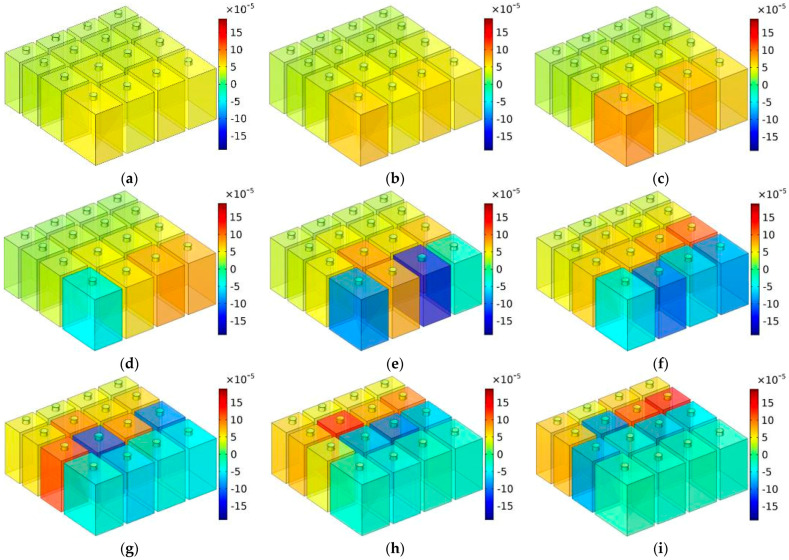
The distributions of TSEDs for MCAM–MNCs–1 with the depth of chamber T = 30 mm: (**a**) 650 Hz; (**b**) 700 Hz; (**c**) 750 Hz; (**d**) 800 Hz; (**e**) 850 Hz; (**f**) 900 Hz; (**g**) 950 Hz; (**h**) 1000 Hz; (**i**) 1050 Hz; (**j**) 1100 Hz; (**k**) 1150 Hz; (**l**) 1200 Hz; (**m**) 1250 Hz; (**n**) 1300 Hz; and (**o**) 1350 Hz.

**Figure 11 materials-16-07627-f011:**
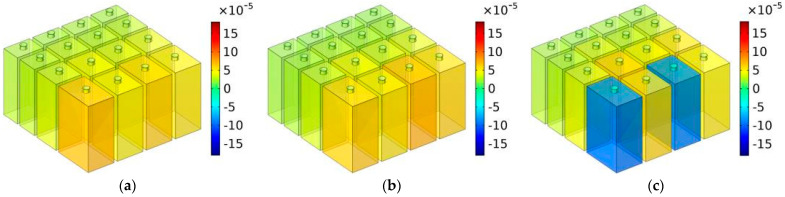
The distributions of TSEDs for MCAM–MNCs–1 with the depth of chamber T = 40 mm: (**a**) 600 Hz; (**b**) 650 Hz; (**c**) 700 Hz; (**d**) 750 Hz; (**e**) 800 Hz; (**f**) 850 Hz; (**g**) 900 Hz; (**h**) 950 Hz; (**i**) 1000 Hz; (**j**) 1050 Hz; (**k**) 1100 Hz; and (**l**) 1150 Hz.

**Figure 12 materials-16-07627-f012:**
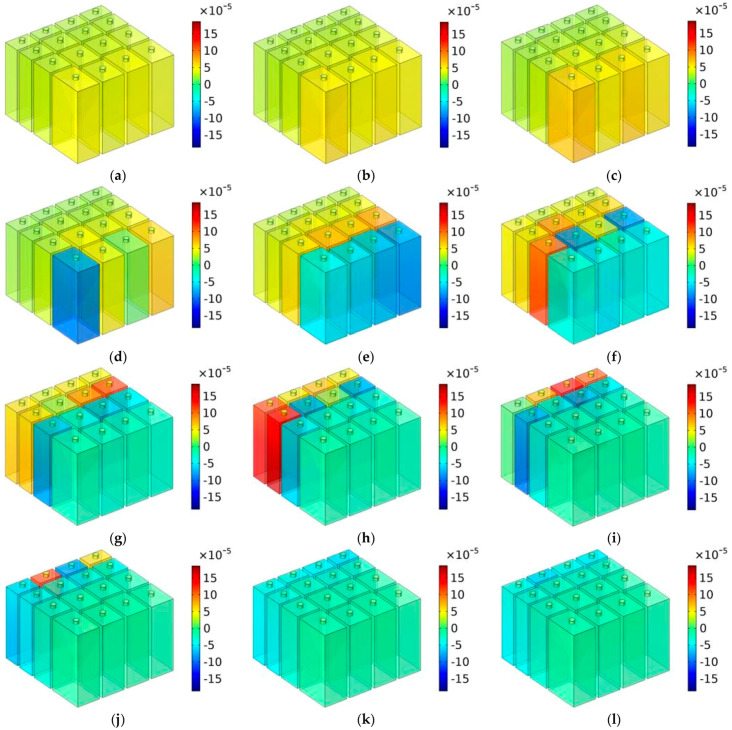
The distributions of TSEDs for MCAM–MNCs–1 with the depth of chamber T = 50 mm: (**a**) 450 Hz; (**b**) 500 Hz; (**c**) 550 Hz; (**d**) 600 Hz; (**e**) 650 Hz; (**f**) 700 Hz; (**g**) 750 Hz; (**h**) 800 Hz; (**i**) 850 Hz; (**j**) 900 Hz; (**k**) 950 Hz; and (**l**) 1000 Hz.

**Figure 13 materials-16-07627-f013:**
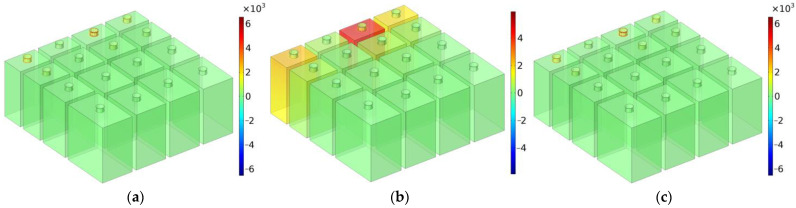
The distributions of acoustic characteristic parameters at the resonance frequency of 1203 Hz for MCAM–MNCs–1 with the depth of rear chamber T = 30 mm: (**a**) viscous power density; (**b**) thermal power density; (**c**) and total power density.

**Table 1 materials-16-07627-t001:** The selected parameters in the acoustic finite element simulation model for MCAM–MNCs [32].

Parameters	Value or Type	Parameters	Value or Type
The type of mesh	Extremely fine mesh	The type of acoustic field	Plane wave
The type of grid	Free tetrahedral grid	The amplitude of the background field	1 Pa
The selected solver	Steady-state solver	The direction of the incident wave	(0, 0, −1)
The maximum unit size	2 mm	The equilibrium pressure	1 atm
The minimum unit size	0.02 mm	The equilibrium temperature	293.15 K
The maximal unit growth rate	1.3	The number of layers in the distribution	15
The curvature factor	0.2	The number of layers in the boundary	8
The resolution of the narrow region	1	The stretch factor in the boundary	1.2
The investigated frequency range	200–1600 Hz	The regulation factor for the thickness	1

**Table 2 materials-16-07627-t002:** The resonance frequencies and peak sound absorption coefficients for the single rectangle Helmholtz resonator with the various lengths of chamber L.

Sound Absorption Performance	The Length of Chamber L (mm)							
	10	12	14	16	18	20	22	24
Resonance frequency (Hz)	1222	1123	1044	1008	977	921	875	835
Peak sound absorption coefficient	0.9947	0.9977	0.9992	0.9998	0.9999	0.9998	0.9992	0.9984

**Table 3 materials-16-07627-t003:** The selected default parameters for the 16 chambers in MCAM–MNCs–1.

Chambers	Thickness of Panel t_0_	Depth of Chamber T_0_	Diameter of Aperture d	Length of Chamber L_i_
C01	2 mm	30 mm	3.5 mm	23.2 mm
C02	2 mm	30 mm	3.5 mm	14.9 mm
C03	2 mm	30 mm	3.5 mm	11.3 mm
C04	2 mm	30 mm	3.5 mm	10.6 mm
C05	2 mm	30 mm	3.5 mm	19.3 mm
C06	2 mm	30 mm	3.5 mm	18.2 mm
C07	2 mm	30 mm	3.5 mm	14.0 mm
C08	2 mm	30 mm	3.5 mm	8.5 mm
C09	2 mm	30 mm	3.5 mm	21.9 mm
C10	2 mm	30 mm	3.5 mm	16.0 mm
C11	2 mm	30 mm	3.5 mm	12.2 mm
C12	2 mm	30 mm	3.5 mm	9.9 mm
C13	2 mm	30 mm	3.5 mm	20.6 mm
C14	2 mm	30 mm	3.5 mm	17.1 mm
C15	2 mm	30 mm	3.5 mm	13.1 mm
C16	2 mm	30 mm	3.5 mm	9.2 mm

**Table 4 materials-16-07627-t004:** The selected default parameters for the 16 chambers in MCAM–MNCs–2.

Chambers	Thickness of Panel t_0_	Depth of Chamber T_0_	Diameter of Aperture d	Length of Chamber L_i_
C01	2 mm	30 mm	3.5 mm	14.2 mm
C02	2 mm	30 mm	3.5 mm	15.8 mm
C03	2 mm	30 mm	3.5 mm	12.7 mm
C04	2 mm	30 mm	3.5 mm	17.3 mm
C05	2 mm	30 mm	3.5 mm	11.3 mm
C06	2 mm	30 mm	3.5 mm	18.7 mm
C07	2 mm	30 mm	3.5 mm	10.0 mm
C08	2 mm	30 mm	3.5 mm	20.0 mm
C09	2 mm	30 mm	3.5 mm	8.8 mm
C10	2 mm	30 mm	3.5 mm	21.2 mm
C11	2 mm	30 mm	3.5 mm	7.7 mm
C12	2 mm	30 mm	3.5 mm	22.3 mm
C13	2 mm	30 mm	3.5 mm	6.7 mm
C14	2 mm	30 mm	3.5 mm	23.3 mm
C15	2 mm	30 mm	3.5 mm	5.8 mm
C16	2 mm	30 mm	3.5 mm	24.2 mm

## Data Availability

The data that support the findings of this study are available from the corresponding author upon reasonable request.

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
