# Peer review of "An Investigation of Modular Composable Acoustic Metamaterials with Multiple Nonunique Chambers"

_materials, 2023, doi:10.3390/ma16247627_

Round 1

Reviewer 1 Report

Comments and Suggestions for Authors

The paper was well written and very clear. I have only one minor comments.

Please add discussion how wide the absorption range can be realized or the limitation of the number of arrayed resonator.

Author Response

Response to reviewer’s comments

General Comment:

The paper was well written and very clear. I have only one minor comments.

Response:

Thank you very much for your helpful review and positive assessment to our manuscript. In this research We have revised the whole manuscript carefully according to your and other reviewers’ comments, and these corrections and modifications are highlighted in yellow in the revised manuscript.

  1. Please add discussion how wide the absorption range can be realized or the limitation of the number of arrayed resonator.

Response:

Thank you very much for your valuable comment and helpful suggestion. The discussions on how wide the absorption range can be realized and the limitation of the number of the arrayed resonators are added respectively at the end of section “3. Parametric Study” in the revised manuscript, and these additions are highlighted in yellow.

Based on the above parametric analysis, it could be found that the width of effective sound absorption range was determined by the structural parameters and the target frequency band together. For example, when the depth of chamber T increased, the gained sound absorption band would shift to the low frequency direction and its width would decrease accordingly, which were consistent with the normal sound absorption principle of Helmholtz resonator. Analogously, if the target frequency band in the low frequency range was desired, the structural parameters should be adjusted accordingly to realize the target and the width of absorption range would decrease as well. Normally speaking, the wide absorption range was easier to obtain in the high frequency range and be difficult to achieve in the low frequency. Moreover, there were total 16 single Helmholtz resonators in the MCAM–MNCs, and the grouping was necessary to gain effective sound absorption. It could be judged from Figures 5–7 that the decrease of group number would enlarge the sound absorption band under the same conditions, but the peak sound absorption coefficients decreased accordingly. Thus, if the uniform sound absorption property in a certain frequency range was needed, the MCAM–MNCs–1 with the group number of 4 was better. Otherwise, it would be better to select the MCAM–MNCs–2 to gain the sound absorption band as wide as possible. The peak sound absorption coefficient and effective sound absorption band should be taken into consideration simultaneously to select the appropriate group numbers for the proposed MCAM–MNCs in this research.

Reviewer 2 Report

Comments and Suggestions for Authors

The paper discusses a metamaterial structure made with resonant cavity systems.

Authors should compare their results with those of other authors.

Metamaterials are new studies in acoustics and therefore the paper is good.

The measurements in the impedance tube confirm that the results reached by the authors are good.

The limits of impedance tube measurements should be added.

Author Response

Response to reviewer’s comments

General Comment: The paper discusses a metamaterial structure made with resonant cavity systems.

Response:

Thank you very much for your kind review to our manuscript and constructive suggestion to our study. We have revised and corrected the whole manuscript carefully according to your and the other reviewers’ comments. The responses to your helpful comments are as follows.

  1. Authors should compare their results with those of other authors.

Response:

Thank you very much for your valuable comment and helpful suggestion. The sound absorption performance of MCAM–MNCs proposed in this research are compared with the other tunable acoustic metamaterials in the literatures. The modifications are added at the end of the section “3. Parametric Study” in the revised manuscript, and the modifications are highlighted in yellow.

Comparing with other tunable acoustic metamaterials in the literatures [19–22], the space utilization rate of MCAM–MNCs was higher, which could obtain better sound absorption property with the limited thickness, because the space division of rear chamber could in-crease the sound absorption efficiency and the front aperture without embedding into the chamber could improve the acoustic flux. Furthermore, the proposed MCAM–MNCs in this research could achieve an adjustable sound absorption performance as well, and the MCAM–MNCs was easier to fabricate and assemble by the modular design, which could reduce the manufacturing cost. These advantages made the MCAM–MNCs more convenient for practical application in the field of noise reduction.

  1. Metamaterials are new studies in acoustics and therefore the paper is good.

Response:

Thank you very much for your helpful review and positive assessment to our research. The MCAM–MNCs was divided to a front panel with same perforated apertures and a rear chamber with nonunique grouped cavities, which made it easy to fabricate and convenient for practical application. The effective sound absorption band was 556 Hz (773–1329 Hz), 456 Hz (646–1102 Hz) and 387 Hz (564–951 Hz) for T=30 mm, T=40 mm and T=50 mm respectively, and the corresponding average sound absorption coefficient was 0.8696, 0.8854 and 0.8916 accordingly, which exhibited excellent sound absorption performance for the MCAM–MNCs. The sound absorption mechanism of the MCAM–MNCs was investigated by the distribution of TSED, which certificated that the sound absorption was obtained by resonance effect of several Helmholtz resonators and the assistance of adjacent Helmholtz resonators.

  1. The measurements in the impedance tube confirm that the results reached by the authors are good.

Response:

Thank you very much for your kind comment. We agree with you that it can be judged from Figure 9b that for the investigated MCAM–MNCs–1, the variation tendencies of the theoretical data, simulation data and experimental data were basically consistent, which could demonstrate the feasibility of the proposed MCAM–MNCs, the reasonability of the constructed theoretical model, and the effectiveness of the selected acoustic finite element simulation method.

  1. The limits of impedance tube measurements should be added.

Response:

Thank you very much for your valuable and meaningful suggestion. The limits of impedance tube measurement were added at the end of the section of “4.2. Standing Wave Tube Testing” in the revised manuscript, and the modifications are highlighted in yellow.

The AWA6290T tester based on the transfer function method could obtain the actual sound absorption coefficients with normal incidence, which were widely applied in the researches on the acoustic metamaterials and the other sound absorbing materials. However, in the practical application, most of the sound wave did not enter the acoustic metamaterial vertically, and random incidence was the normal situation. Thus, the reverberation chamber method to test the sound absorption performance was closer to real condition, although it needed more samples and more time. Normally speaking, the sample of acoustic metamaterial was measured through the standing wave tube in the laboratory at the research stage, and it was further tested in the reverberation chamber before the practical application.

Reviewer 3 Report

Comments and Suggestions for Authors

The paper introduces a novel approach, Modular Composable Acoustic Metamaterial with Multiple Nonunique Chambers (MCAM–MNCs), designed for easy fabrication and practical application in sound absorption. The structure is divided into a front panel with identical perforated apertures and a rear chamber with nonunique grouped cavities. The study employs acoustic finite element simulation to conduct parametric studies on the diameter of the aperture (d), depth of chamber (T0), and thickness of the panel (t0). The results demonstrate the tunability of sound absorption performances for MCAM–MNCs–1 and MCAM–MNCs–2, showcasing promising noise reduction effects.

The proposed MCAM–MNCs structure is innovative and presents a novel approach to sound absorption. The modular and composable design and nonunique chambers add versatility to the absorber, making it potentially suitable for various applications.

The parametric studies on the diameter of the aperture, the chamber's depth, and the panel's thickness are well-conducted. The results provide valuable insights into the impact of these parameters on the sound absorption performance, enhancing the understanding of the material behavior.

The acoustic finite element simulation results are comprehensive and well-presented. The effective sound absorption band and corresponding average sound absorption coefficients for different configurations are clearly outlined, demonstrating the efficacy of the proposed MCAM–MNCs structure.

The investigation into the sound absorption mechanism, mainly through the distributions of total sound energy density (TSED), adds depth to understanding how MCAM–MNCs achieve its noise attenuation performance. This section could be expanded further to provide a more detailed analysis.

The use of additive manufacturing to fabricate components for MCAM–MNCs is commendable. However, the sound absorption coefficients should be discussed in greater detail, including any deviations from simulated results and potential challenges during the experimental process.

 Overall, the paper is well-written and organized. The figures and tables effectively support the content. However, a more detailed discussion of the practical application of MCAM–MNCs and potential real-world scenarios would enhance the paper's impact and relevance.

The paper introduces an intriguing concept with MCAM–MNCs for modular composable acoustic metamaterials. The thorough simulation results, parametric studies, and experimental validation through additive manufacturing contribute significantly to the field. Addressing the key comments above will strengthen the paper and its potential impact. Some key modular acosutc absorbers related papers can be cited: 10.1063/5.013985610.1016/j.eml.2020.100657

Minor revisions are suggested to improve the sound absorption mechanism's clarity and provide a more comprehensive discussion on the practical application of MCAM–MNCs.

The paper shows a high level of novelty and technical rigor, so it's considered for publication after addressing the revisions.

Author Response

Response to reviewer’s comments

General Comment:

The paper introduces a novel approach, Modular Composable Acoustic Metamaterial with Multiple Nonunique Chambers (MCAM–MNCs), designed for easy fabrication and practical application in sound absorption. The structure is divided into a front panel with identical perforated apertures and a rear chamber with nonunique grouped cavities. The study employs acoustic finite element simulation to conduct parametric studies on the diameter of the aperture (d), depth of chamber (T0), and thickness of the panel (t0). The results demonstrate the tunability of sound absorption performances for MCAM–MNCs–1 and MCAM–MNCs–2, showcasing promising noise reduction effects.

Response:

Thank you very much for your helpful review and positive assessment to our manuscript. We have revised the whole manuscript carefully according to your and other reviewers’ comments, and these corrections and modifications are highlighted in yellow in the revised manuscript.

  1. The proposed MCAM–MNCs structure is innovative and presents a novel approach to sound absorption. The modular and composable design and nonunique chambers add versatility to the absorber, making it potentially suitable for various applications.

Response:

Thank you very much for your kind comment. In order to better explain the advantages of the proposed MCAM–MNCs structure, some presentations are added according to other reviewers’ comments.

The proposed MCAM–MNCs in this research could achieve an adjustable sound absorption performance as well, and the MCAM–MNCs was easier to fabricate and assemble by the modular design, which could reduce the manufacturing cost. These advantages made the MCAM–MNCs more convenient for the practical application in the field of noise reduction.

These presentations are added at the end of the section “3. Parametric Study” in the revised manuscript, and they are highlighted in yellow.

  1. The parametric studies on the diameter of the aperture, the chamber's depth, and the panel's thickness are well-conducted. The results provide valuable insights into the impact of these parameters on the sound absorption performance, enhancing the understanding of the material behavior.

Response:

Thank you very much for your kind comment. To better show the parameters’ effect, some presentations are added according to other reviewers’ comments.

Based on the above parametric analysis, it could be found that the width of effective sound absorption range was determined by the structural parameters and the target frequency band together. For example, when the depth of chamber T increased, the gained sound absorption band would shift to the low frequency direction and its width would decrease accordingly, which were consistent with the normal sound absorption principle of Helmholtz resonator. Analogously, if the target frequency band in the low frequency range was desired, the structural parameters should be adjusted accordingly to realize the target and the width of absorption range would decrease as well. Normally speaking, the wide absorption range was easier to obtain in the high frequency range and be difficult to achieve in the low frequency. Moreover, there were total 16 single Helmholtz resonators in the MCAM–MNCs, and the grouping was necessary to gain effective sound absorption. It could be judged from Figures 5–7 that the decrease of group number would enlarge the sound absorption band under the same conditions, but the peak sound absorption coefficients decreased accordingly. Thus, if the uniform sound absorption property in a certain frequency range was needed, the MCAM–MNCs–1 with the group number of 4 was better. Otherwise, it would be better to select the MCAM–MNCs–2 to gain the sound absorption band as wide as possible. The peak sound absorption coefficient and effective sound absorption band should be taken into consideration simultaneously to select the appropriate group numbers for the proposed MCAM–MNCs in this research.

These presentations are added at the end of the section “3. Parametric Study” in the revised manuscript, and they are highlighted in yellow.

  1. The acoustic finite element simulation results are comprehensive and well-presented. The effective sound absorption band and corresponding average sound absorption coefficients for different configurations are clearly outlined, demonstrating the efficacy of the proposed MCAM–MNCs structure.

Response:

Thank you very much for your kind comment. The acoustic finite element simulation method is utilized in this study, which aims to improve the research efficiency, decrease the research cost, and exhibit the sound absorption mechanism.

  1. The investigation into the sound absorption mechanism, mainly through the distributions of total sound energy density (TSED), adds depth to understanding how MCAM–MNCs achieve its noise attenuation performance. This section could be expanded further to provide a more detailed analysis.

Response:

Thank you very much for your helpful suggestion. In the original manuscript, the sound absorption mechanism is discussed by the distribution of TSED at the investigated frequency points in the certain range with the interval of 50 Hz. According to your and other reviewers’ comments, the distributions of acoustic characteristic parameters at the resonance frequency were analyzed as well, which could better explore the sound absorption mechanism of MCAM–MNCs. The distributions of viscous power density, thermal power density and total power density at the resonance frequency of 1203 Hz for the MCAM–MNCs–1 with the depth of chamber T=30 mm are shown in the Figure 13. It could be found that the sound absorption in MCAM–MNCs was realized by the thermal viscosity effect, and the viscous energy loss in the front aperture was the dominant factor relative to the thermal energy loss in the rear chamber, because the value of former was thousands of times the latter, which could be judged from the comparisons of Figures 13a and 13b.

(a)

(b)

(c)

Figure 13. The distributions of acoustic characteristic parameters at the resonance frequency of 1203 Hz for the MCAM–MNCs–1 with the depth of rear chamber T=30 mm. (a) viscous power density; (b) thermal power density; (c) and total power density.

  1. The use of additive manufacturing to fabricate components for MCAM–MNCs is commendable. However, the sound absorption coefficients should be discussed in greater detail, including any deviations from simulated results and potential challenges during the experimental process.

Response:

Thank you very much for your kind inquiry and valuable suggestion. According to your and other reviewers’ comments, the section “4.2. Standing Wave Tube Testing” is enriched and further discussed in the revised manuscript, and this correction is highlighted in yellow.

The AWA6290T tester based on the transfer function method could obtain the actual sound absorption coefficients with a normal incidence, which were widely applied in the researches on the acoustic metamaterials and other sound absorbing materials. However, in the practical application, most of the sound wave did not enter the acoustic metamaterial vertically, and the random incidence was the normal situation. Thus, the reverberation chamber method to test the sound absorption performance was closer to the real condition, although it needed more samples and more time. Normally, the sample of acoustic metamaterial was measured through the standing wave tube in the laboratory at the re-search stage, and it was further tested in the reverberation chamber before practical application.

  1. Overall, the paper is well-written and organized. The figures and tables effectively support the content. However, a more detailed discussion of the practical application of MCAM–MNCs and potential real-world scenarios would enhance the paper's impact and relevance.

Response:

Thank you very much for your positive assessment to our study and helpful suggestion to the manuscript. The potential applications of MCAM–MNCs in real-world scenarios are added at the end of section “6. Conclusions”, and they are highlighted in yellow in the revised manuscript.

The proposed MCAM–MNCs has the advantages of excellent sound absorption performance, adjustable noise reduction frequency band, easy to manufacture, convenient practical application, and so on, which make it the potential sound absorber to control the low frequency variational noise in the actual scenarios.

  1. The paper introduces an intriguing concept with MCAM–MNCs for modular composable acoustic metamaterials. The thorough simulation results, parametric studies, and experimental validation through additive manufacturing contribute significantly to the field. Addressing the key comments above will strengthen the paper and its potential impact.  Some key modular acosutc absorbers related papers can be cited: 10.1063/5.0139856, 10.1016/j.eml.2020.100657

Response:

Thank you very much for your meaningful suggestion. We have read the two recommended literatures carefully and utilized them to replace the original references [13] and [15].

[1] Kumara, S.; Lee, H.P. Reconfigurable metatiles with circular maze-like space-coiling-based acoustic metastructure for low-to-mid frequency sound attenuation. J. Appl. Phys. 2023, 133, 154901.

[2] Wu, L.; Liu, L.; Wang, Y.; Zhai, Z.; Zhuang, H.; Krishnaraju, D.; Wang, Q.; Jiang, H. A machine learning-based method to design modular metamaterials. Extrem. Mech. Lett. 2020, 36, 100657.

  1. Minor revisions are suggested to improve the sound absorption mechanism's clarity and provide a more comprehensive discussion on the practical application of MCAM–MNCs.

Response:

Thank you very much for your helpful suggestion.

To improve the sound absorption mechanism's clarity, the distributions of acoustic characteristic parameters at the resonance frequency were analyzed as well, which could better explore the sound absorption mechanism of MCAM–MNCs. This part is same with the response to your review comment 4.

To provide a more comprehensive discussion on the practical application of MCAM–MNCs, the potential applications of MCAM–MNCs in the real-world scenarios are added at the end of section “6. Conclusions”. This part is same with the response to your review comment 6.

  1. The paper shows a high level of novelty and technical rigor, so it's considered for publication after addressing the revisions.

Response:

Thank you very much again for your helpful review and positive assessment to our manuscript. We have revised the whole manuscript carefully according to your and other reviewers’ comments, and these corrections and modifications are highlighted in yellow in the revised manuscript.

Reviewer 4 Report

Comments and Suggestions for Authors

Well written paper.

in paragraph 4, you will discuss impedance tube measurements, they are normal incidence measurements. The specimens must be inserted into the tube, the ratio of the wavelength to the dimensions of the specimens must always be considered.

Increase the references by recalling the first studies on this topic by Veselago, Pendry. To date many studies have been performed by Alu. you also see that Trematerra used 3D systems and Bevilacqua 2D systems of metamaterials

Author Response

Response to reviewer’s comments

General Comment:

Well written paper.

Response:

Thank you very much for your helpful review and positive assessment to our manuscript. We have revised the whole manuscript carefully according to your and the other reviewers’ comments, and these corrections and modifications are highlighted in yellow in the revised manuscript.

  1. in paragraph 4, you will discuss impedance tube measurements, they are normal incidence measurements. The specimens must be inserted into the tube, the ratio of the wavelength to the dimensions of the specimens must always be considered.

Response:

Many thanks for your valuable comment and helpful suggestion. According to your comment, the following presentations are corrected in the section of “4.2. Standing Wave Tube Testing” in the revised manuscript, and these corrections are highlighted in yellow.

The tested MCAM–MNCs–1 sample was fixed in the sample tube and its front surface was next to the end of standing wave tube.

the distance between the 2 microphones and that between microphone 2 and the tested MCAM–MNCs–1 sample were 70 mm and 170 mm respectively, which could test the sound absorption coefficients in the frequency range of 200–1600 Hz.

the first sound absorption peak was gained at the resonance frequency around 830 Hz, which meant that the total thickness of MCAM–MNCs–1 sample 34 mm was only 1/12 of sound wavelength 409 mm (λ=c/f=340/830×1000=409 mm). It could be proved that the MCAM–MNCs could obtain high absorption efficiency and wide absorption band simultaneously.

The AWA6290T tester based on the transfer function method could obtain the actual sound absorption coefficients with a normal incidence, which were widely applied in the researches on the acoustic metamaterials and other sound absorbing materials.

the sample of acoustic metamaterial was measured through the standing wave tube in the laboratory at the research stage, and it was further tested in the reverberation chamber be-fore practical application.

  1. Increase the references by recalling the first studies on this topic by Veselago, Pendry. To date many studies have been performed by Alu. you also see that Trematerra used 3D systems and Bevilacqua 2D systems of metamaterials.

Response:

Thank you very much for your valuable comment and helpful suggestion. We have sought the names of these provided authors (Veselago; Pendry; Alu; Trematerra; Bevilacqua) with the topic of metamaterial in the web of science, and the following literatures are found.

(1) For Veselago, only 2 literatures are found, but it doesn’t seem to have any relationship with acoustic metamaterial for sound absorption. Thus, these 2 literatures are added to the reference list in the revised manuscript.

(2) For Pendry, there are total 28 literatures found, and 1 of them are selected to replace the original reference [38].

[1] Pendry, J.; Zhou, J.; Sun, J.B. Metamaterials: From Engineered Materials to Engineering Materials. Engineering 2022, 17, 1–2.

(3) For Alu, there are total 138 literatures found, and 1 of them are selected to replace the original reference [30].

[1] Askarpour, A.; Zhao, Y.; Alu, A. Wave propagation in twisted metamaterials. Phys. Rev. Condens. Matter Mater. Phys. 2014, 90, 054305.

(4) For Trematerra and Bevilacqua, there are 3 and 2 literatures on acoustic metamaterials found respectively, and they are in one research team. Thus, 2 literatures are selected to replace the original references [4] and [6].

[1] Trematerra, A.; Bevilacqua, A.; Iannace, G. Noise Control in Air Mechanical Ventilation Systems with Three-Dimensional Metamaterials. Appl. Sci. 2023, 13, 1650.

[2] Bevilacqua, A.; Iannace, G.; Lombardi, I.; Trematerra, A. 2D Sonic Acoustic Barrier Composed of Multiple-Row Cylindrical Scatterers: Analysis with 1:10 Scaled Wooden Models. Appl. Sci. 2022, 12, 6302.
